# Wound Healing and Cell Dynamics Including Mesenchymal and Dental Pulp Stem Cells Induced by Photobiomodulation Therapy: An Example of Socket-Preserving Effects after Tooth Extraction in Rats and a Literature Review

**DOI:** 10.3390/ijms21186850

**Published:** 2020-09-18

**Authors:** Yuki Daigo, Erina Daigo, Hiroshi Fukuoka, Nobuko Fukuoka, Masatsugu Ishikawa, Kazuya Takahashi

**Affiliations:** 1Department of Geriatric Dentistry, Osaka Dental University, Osaka 546-0032, Japan; kazuya-t@cc.osaka-dent.ac.jp; 2Department of Anesthesiology, Osaka Dental University, Osaka 540-0008, Japan; daigou_e0120@yahoo.co.jp; 3Fukuoka Dental Office, Satsuma-gun, Kagoshima 895-1811, Japan; hn-fukuoka@po4.synapse.ne.jp (H.F.); nobuko-aoki@hotmail.co.jp (N.F.); 4Bees Dental Office, Fukuoka City, Fukuoka 814-0106, Japan; beezaemonn@yahoo.co.jp

**Keywords:** CO_2_ laser, diode laser, tooth extraction, socket preservation, extraction socket, photobiomodulation therapy, high-intensity laser therapy, wound healing

## Abstract

High-intensity laser therapy (HILT) and photobiomodulation therapy (PBMT) are two types of laser treatment. According to recent clinical reports, PBMT promotes wound healing after trauma or surgery. In addition, basic research has revealed that cell differentiation, proliferation, and activity and subsequent tissue activation and wound healing can be promoted. However, many points remain unclear regarding the mechanisms for wound healing induced by PBMT. Therefore, in this review, we present an example from our study of HILT and PBMT irradiation of tooth extraction wounds using two types of lasers with different characteristics (diode laser and carbon dioxide laser). Then, the effects of PBMT on the wound healing of bone tissues are reviewed from histological, biochemical, and cytological perspectives on the basis of our own study of the extraction socket as well as studies by other researchers. Furthermore, we consider the feasibility of treatment in which PBMT irradiation is applied to stem cells including dental pulp stem cells, the theme of this Special Issue, and we discuss research that has been reported on its effect.

## 1. Introduction

Medical lasers have three key elements, namely, thermal properties such as power, optical properties such as wavelength, and operating mode such as pulsed or continuous wave, and can be divided into two types depending on differences in the depth of tissue penetration (tissue transmission and tissue surface absorption). The therapeutic effects of medical lasers depend on synergistic interactions of these three elements and related characteristics. High-intensity laser therapy (HILT), the parameters of which are above the survival threshold, causes an irreversible reaction, and thus is used for tissue incision and vaporization, whereas photobiomodulation therapy (PBMT), the parameters of which are within the survival threshold, causes a reversible reaction. PBMT exerts photobiological and photochemical actions that stimulate tissues to induce an anti-inflammatory response and wound healing [1,2,3], pain relief [4,5], cell division and proliferation [6,7], and inhibiting scar formation [8,9,10,11]. The type of laser and the irradiation conditions used vary according to the purpose, the patient, and the expected effects of laser treatment.

We have investigated wound healing in the extraction socket (a recess formed after tooth extraction) following laser irradiation for the past 10 years; a summary of our research results is provided in Section 4 [12,13]. First, the importance of wound healing of the extraction socket is briefly explained.

The height of the alveolar crest is highly likely to be reduced after tooth extraction, leaving a considerable recess in the alveolar mucosa. Maintaining the condition of the alveolar bone and preserving alveolar crest height are crucial for long-term maintenance of functioning prostheses such as dentures, crowns, and implants. In particular, implant treatment is said to require an alveolar bone height of 10 mm or more and a width of 6 mm or more. However, these requirements depend on the implant type, length, width, and superstructure type. Furthermore, the quality of the alveolar bone varies widely from person to person, and thus it is not possible to specify the precise requirements. In addition, the requirements cannot be specified internationally because the guidelines differ by country. For these reasons, preserving alveolar crest height, together with rapid wound healing after tooth extraction, is essential. This is the concept behind socket preservation.

Currently, the standard socket preservation techniques use autologous grafts (e.g., those harvested from the ilium or mandibular bone) or artificial fillers (bone filler or collagen sponge). These materials promote blood retention and bone regeneration within the extraction socket, although there are some problems associated with their use. For example, there is a risk of graft-associated infection, it takes a long time for the grafted site to be replaced by bone tissue, some residual graft materials may remain and not be replaced by bone tissue, and the procedure must be performed by a highly skilled surgeon. For these reasons, socket preservation by laser irradiation has attracted great interest as an alternative technique.

The guidelines for dental laser treatment proposed by the U.S. Food and Drug Administration (FDA) in 2009 included “coagulation of extraction sites” using CO_2_ lasers and diode lasers [14]. Favorable wound healing has been demonstrated using both types of lasers in clinical cases, but most of the basic research conducted to date has mostly investigated diode lasers, with research on CO_2_ lasers remaining extremely limited.

This article reviews the characteristics of CO_2_ and diode lasers, which are two types of laser equipment recommended by the FDA, the clinical application of laser irradiation based on those characteristics, and the findings of basic research on the acceleration of wound healing and its cell dynamics in the extraction socket as an example, including an account of our previous research.

## 2. Characteristics of CO_2_ and Diode Lasers

CO_2_ lasers have the longest wavelength (10.6 µm) among dental lasers. For comparison, other dental lasers with shorter wavelengths include the Er:YAG laser (2940 nm), followed by the Nd:YAG laser (1064 nm), diode laser (600–1000 nm), and argon laser (488 nm). The energy of CO_2_ lasers is more readily absorbed by water compared with other lasers. During CO_2_ laser irradiation treatment, most of the laser’s energy is absorbed and converted into heat in the skin tissue about 0.05 mm from the surface; therefore, CO_2_ lasers are classified as tissue surface absorption lasers. In contrast, diode lasers are classified as tissue transmission lasers because they have a short wavelength (approximately 600–1000 nm for dental lasers) and their energy is not easily absorbed by water but is transmitted deep into the tissue instead reaching the subcutaneous tissue of the skin.

## 3. Clinical Application and Basic Research on CO_2_ Laser and Diode Laser Irradiation of Tooth Extraction Sockets

To minimize changes in the alveolar bone after tooth extraction, a blood clot should be retained and the wound surface should be protected at the early stage; this will prevent mucosal and epithelial concavity from forming and will promote new bone formation. All of this can be achieved using CO_2_ lasers and diode lasers. A laser treatment protocol used for clinical cases is shown in Figure 1.

Once blood has completely filled the tooth extraction socket, HILT is performed to coagulate and carbonize the surface of the blood and to fuse the resulting eschar to the surrounding gingiva to prevent dislodgement. PBMT is performed the following day to activate the wound tissue and enhance would healing.

Some case reports [15] have shown good preservation of alveolar height by using a CO_2_ dental laser after tooth extraction, reflecting the experience of many dentists in the clinical setting. However, there is considerable skepticism due to lack of adequate supportive evidence from basic research.

Therefore, in our previous research [12,13], which we present below, we conducted basic research to investigate the wound healing effect of CO_2_ laser and diode laser irradiation after tooth extraction and the underlying cell dynamics.

## 4. Observation of Wound Healing in the Tooth Extraction Socket Treated Using a CO_2_ Laser or Diode Laser

This section describes the methods used in our recent study [12]. Five-week-old male Wistar rats (weight, 130–150 g) were used for the experiment. The rats were housed three per cage with free access to food (CLEA Rodent Diet CE-2; CLEA Japan, Inc.) and tap water. The room was maintained at 24 ± 2 °C and 50 ± 5% humidity with 12-h light/dark cycles. The rats were divided into two groups according to post-extraction treatment: combined HILT and PBMT using a CO_2_ laser (CO_2_ group), a diode laser (diode group), and no laser irradiation (control group). The term ‘‘laser treatment groups’’ is used to describe the diode and CO_2_ groups collectively. Observation time points were post-extraction days 3, 7, and 21. A total of 54 rats, six for each observation time point, were examined in this study.

The surgical procedures for each group were as follows. Rats were anesthetized by intraperitoneal administration of pentobarbital sodium and subjected to extraction of the left maxillary first molar using a rat-use elevator and mosquito forceps. Following this, the procedure was performed as shown in Figure 1.

Details of the lasers and irradiation conditions used are shown below.

The CO_2_ laser was a PanalasCO5Σ (Panasonic Shikoku Electronics Co., Ltd., Osaka, Japan) with the following specifications: wavelength, 10,600 nm; laser tip inner diameter, 0.15 cm (Taper 1A; transmittance 90%). The irradiation conditions were as follows.

(1)HILT: Irradiation was performed with the laser tip not in contact with the blood on the surface of the socket (1.0 W, continuous-wave mode, non-air, 30 s, 27 J).(2)PBMT: Irradiation was performed with the laser tip slightly in contact with the scab on the surface of the socket (1.0 W, Σ-mode, non-air, 15 s, 0.7 J). Σ-mode uses an ultra-short pulse width to increase peak power during irradiation, thereby enabling photobiomodulation (pulse time = 0.0008 s, pulse interval = 0.03 s, 1 cycle = 0.0308 s).

The diode laser was an iLase (Biolase Technology, Inc., Irvine, CA, USA), with the following specifications: wavelength, 940 nm; fiber laser tip; spot diameter, 0.4 mm.

(1)HILT: Irradiation was performed with the laser tip placed slightly in contact with blood on the surface of the socket (1.0 W, continuous-wave mode, 27 s, 27 J).(2)PBMT: Irradiation was performed with the laser tip placed slightly in contact with the scab on the surface of the socket (0.3 W, CP1-mode, 7 s, 0.7 J). The CP1-mode is a gated pulse mode (pulse duration, 0.0001 s; pulse interval, 0.0002 s; 1 cycle, 0.0003 s).

### 4.1. Histopathological Analysis

The extraction socket with the surrounding tissue was removed en bloc from each rat after a lethal dose of anesthetic and was fixed in 4% paraformaldehyde for 48 h. After a 3-week decalcification in 10% ethylenediaminetetraacetic acid followed by dehydration using an alcohol gradient, tissue samples were embedded in paraffin. Consecutive sagittal sections (4 µm thick) were prepared using a microtome and subjected to hematoxylin–eosin staining, after which histopathological observation was performed.

Images were scanned using a digital microscope (VZ-9000; Keyence Co., Ltd., Osaka, Japan) and analyzed using Scionimage software (Scion Corporation, Frederick, MD, USA).

The study protocol complied with the Osaka Dental University Guidelines for Animal Experiments (approval no. 18-01008).

### 4.2. Results

Healing of a tooth extraction socket is a four-stage process involving coagulation, granulation tissue formation, temporary bone formation, and healing. The last three stages are described below.

### 4.3. Post-Extraction Day 3 (Corresponds to the Granulation Tissue Stage in Humans)

Sockets were almost completely filled with a blood clot in the control group and the progression of organization from the socket walls was observed in the laser treatment groups (Figure 2a–c). In particular, organization from the socket fundus was observed to be rapid in the diode group. Magnified views of socket walls showed few osteoclast-like cells in the control group; however, many osteoclast-like cells were observed in the CO_2_ group. Other findings indicated active bone resorption in the laser treatment groups (Figure 2d,e). In this review, osteoclast-like cells were defined as a set of multinucleated giant cells present on the surface of the tooth extraction wall and the bone-resorbing cavities in contact with these cells.

### 4.4. Post-Extraction Day 7 (Corresponds to the Temporary Bone Stage in Humans)

Simultaneous bone resorption by osteoclast-like cells and bone formation around the socket were observed in the control group, whereas few osteoclast-like cells were observed in the laser treatment groups. During new bone formation in the control group, there was less cancellous bone and the cancellous bone that had formed was immature. In the CO_2_ group, new bone formation was confirmed from the middle to the shallow layer with a cross-linking pattern and the trabeculae were dense and wide. New bone formation was confirmed from the socket fundus to the shallow layer of the socket in the diode group (Figure 2g–i).

### 4.5. Post-Extraction Day 21 (Corresponds to the Healing Stage in Humans)

Extraction sockets were filled with mature new bone with dense cancellous bone in all groups. However, a concavity was noted at the center of the alveolar crest in the control group, whereas no concavity was observed in the laser treatment groups (Figure 2j–l).

## 5. Comparison with the Current Socket Preservation Method

Various approaches for suppressing alveolar bone resorption after tooth extraction have been investigated in basic and clinical research, many of which involve socket grafting with autologous bone or artificial bone material. In dental clinical research, Araùjo et al. measured the bone area on cross-sectional CT images of extraction sockets, and found only an approximately 3% reduction in a group that received Bio-Oss Collagen (Geistlich Pharma AG., Wolhusen, Switzerland) followed by palatal mucosa grafting, compared with an approximately 25% reduction in a control group [16]. Iasella et al. reported alveolar bone resorption in the horizontal direction of 2.6 ± 2.3 mm in a control group compared with 1.2 ± 0.9 mm in a group that received freeze-dried bone graft in the extraction socket [17]. Avila et al. reviewed several clinical studies including the above and reported favorable preservation of alveolar bone height by socket grafting using autologous bone and artificial bone material [18]. However, Chan et al. reported that residual (15–36%) grafting materials such as Bio-Oss Collagen and hydroxyapatite remained in granular form for up to 6 months after placement in the extraction socket [19]. Furthermore, Esposito et al. showed that the duration required for healing after bone augmentation was 3 months longer with Bio-Oss Collagen than with autologous bone [20]. The guided bone regeneration (GBR) method uses a membrane to prevent the invasion of the gingiva, that is, to secure a site for new bone formation by making space and promoting regeneration. In some cases, autologous bone, artificial bone, or a substitute material is transplanted. A systematic review of the effects of barrier membranes on bone formation reported an average increase in vertical bone formation of 0.32 mm, minimal postoperative infection, minimal wound dehiscence, and minimal exposure of membranes and implants [21]. However, Chen et al. reported that immediate implant placement, which is performed simultaneously with the GBR method, may cause the membrane to rupture or be exposed [22]. The platelet-rich plasma method, which fills the extraction socket with the collected and separated autologous plasma, is also attracting increased attention. However, because the substance to be filled is a blood component, it is necessary to mix it with an artificial material or shield it with a membrane, rather than using it alone [23]. Taken together, socket preservation with autologous bone and artificial bone material suppresses alveolar resorption after tooth extraction in the horizontal and vertical directions, although there are problems such as delayed wound healing and increased risk graft-associated complications.

In basic research thus far, George et al. [24] and Hisanaga et al. [25] previously showed that a collagen sponge placed in the extraction socket enhanced differentiation and proliferation of mesenchymal stem cells (MSCs) and also served as a scaffold upon which new bone was formed. In addition, Mendes et al. studied the application of sodium hyaluronate, which maintains tissue homeostasis and accelerates osteoinduction, in extraction sockets of rats, showing active alveolar bone resorption on post-extraction day 7, followed by marked new bone formation; on post-extraction day 21, bone density was high and the border between the preexisting alveolar bone and the newly formed bone was blurred [26].

In our work, when the extraction socket was treated with a combination of HILT and PBMT using a CO_2_ laser, we observed many osteoclast-like cells in the relatively shallow layers (i.e., shallow to middle layers) of the socket walls and active bone resorption (activation of bone remodeling) was indicated in the same layers on post-extraction day 3; new bone formation with a unique cross-linking pattern and maturation of trabeculae were observed in the same layers on post-extraction day 7, and the center of the alveolar crest was flat, and no concavity was observed on post-extraction day 21. Meanwhile, in the diode group, organization from the whole circumference of the socket walls, which was more advanced compared with that in the CO_2_ group, was observed on post-extraction day 3. Cancellous bone occupying two-thirds of the socket was observed on post-extraction day 7. On post-extraction day 21, as in the CO_2_ group, new bone formed to a height similar to the neighboring alveolar crests, leaving no concavity in the surface on the healed socket.

These morphological findings and the course of the healing process are in good agreement with findings in previous research.

## 6. Mechanism of the Socket-Preserving Effect Using a CO_2_ Laser or Diode Laser

HILT and PBMT using a CO_2_ laser after tooth extraction are important for socket preservation [12,13]. One reason why the alveolar bone height was preserved well without formation of a concavity in our research was that the surface of the blood was rapidly carbonized to form an artificial eschar. This allowed the blood clot to be retained at a high position in the extraction socket, preventing a mucosal epithelial concavity. As a result, new bone with a good height was formed in the extraction socket (space-making effect of an artificial eschar in new bone formation). This is in good agreement with a study by Huebsch and Hansen, which found that the position of the blood clot in the extraction socket determined the height of the newly formed bone [27]. This also indicates that loss of alveolar height occurs when the extraction socket is not adequately filled with blood or when the blood volume in the extraction socket is not maintained, thereby allowing mucosal an epithelial concavity to form.

The enhanced wound healing shown in our research is likely attributable to wound tissue activation and the interaction of light with bone tissue (and cells involved in bone formation) during PBMT, which is in good agreement with the previous studies described earlier. New bone formation is usually initiated from the floor of the extraction socket. In addition, new bone formation with a cross-linking pattern was confirmed from the shallow to middle layers in the CO_2_ group. Given the extremely shallow light penetration of a CO_2_ laser (0.05 mm), the location where photobioactive reactions are likely to occur is in the shallow to middle layers of the extraction socket. In other words, the location of new bone formation with a cross-linking pattern matched the location of photobioactive reactivity. On the other hand, unlike the CO_2_ laser, the diode laser showed photobioactive reactivity in deep tissue over a wide area, resulting in the formation of new bone in about two-thirds of the extracted tooth socket in the early stage of wound healing of bone tissue.

Taken together, the findings of our research and previous basic research studies suggest the following process may occur during socket preservation by CO_2_ laser or diode laser irradiation (Figure 3). The wound healing mechanism of the extraction socket by CO_2_ laser irradiation is considered as follows. The photobioactive reaction in the shallow to middle layers of the extraction socket wall during PBMT using a CO_2_ laser stimulates undifferentiated MSCs, bone marrow stem cells, osteoblasts, osteoclasts, and bone lining cells in these layers, leading to enhancement of cell differentiation and proliferation followed by bone remodeling. Osteoblasts are stimulated to produce collagen fibers and proteins associated with bone formation. The collagen fibers form a scaffold upon which the osteoblasts migrate to form new bone. As a result, new bone with a characteristic cross-linked pattern is formed in these layers, thereby preventing a mucosal epithelial concavity and forming a flat surface on the alveolar crest. In addition, wound healing promoted by irradiation with a diode laser is considered as follows. The photobioactive reaction induced by PBMT using a diode laser occurs at the depth within the tissue to which the light penetrates, stimulating all osteogenesis-associated cells to accelerate new bone formation. As a result, organization occurs in the socket at the early stage of treatment, and then fibrous granulation tissue is formed in the entire socket to serve as a scaffold for new bone formation. Taken together, although the healing mechanisms vary depending on the laser characteristics, PBMT is expected to activate osteogenesis-associated cells in the area to which the laser can penetrate, thereby enhancing wound healing.

Regarding potential problems in socket preservation using CO_2_ laser and diode laser irradiation, we consider there to be basically no side effects like those of drugs. However, errors may arise due to incorrectly set irradiation conditions or mistakes in the irradiation method. In particular, when using a diode laser, there is a high risk of the laser tip coming into contact with the bone surface because the laser tip must be brought into contact with the blood. As a result, there is a risk of burns and severe pain in the bone tissue, or worse, osteonecrosis. In addition, electricity is used to generate diode lasers, and thus caution is required for patients with pacemakers or artificial valves. In contrast, the laser tip of a CO_2_ laser does not require contact with the blood, and thus there is little or no effect on bone tissue, making it the safer option.

Laser irradiation of the extraction socket can be applied to nearly every patient. This method would be particularly effective in patients taking anticoagulants or antithrombotic agents or those prone to bleeding.

## 7. Effects of Laser Irradiation on Bone Tissues, Including Tooth Extraction Sockets, and the Underlying Cell Dynamics

We next discuss recent basic research on PBMT and osteogenesis, which has elucidated the mechanism of wound healing in the extraction socket in PBMT. Although there have been few reports on the wound healing of bone tissue with PBMT using a CO_2_ laser, it has been verified. Tang and Chai showed that PBMT using a CO_2_ laser on experimental radial fractures in rats resulted in the appearance of many osteoclasts, increased blood supply due to increased capillary formation, and accelerated calcification at the early stage of treatment [28]. In addition, Tsai et al. showed that PBMT enhanced healing of artificially damaged rabbit menisci via a significant increase in the proliferation of fibroblasts and a marked increase in the amount of collagen fibers [29]. Nevertheless, basic research on the effect of CO_2_ laser irradiation on the bone tissue remains limited, and many questions remain unanswered.

In contrast, the diode laser has been well studied. Histopathology revealed the appearance and organization of many fibroblasts at the early healing stage, followed by enhanced new bone formation, acceleration of trabecular maturation, and a significant increase in bone volume [30,31]. In addition, biochemistry revealed significant increases in osteocalcin secretion by osteoblasts [32,33] as well as type I collagen expression [34], suggesting an enhancing effect on would healing. Stein et al. showed that during bone tissue healing, alkaline phosphatase activity and the expression of type I collagen and osteopontin increases within 72 h after irradiation [35], and Saracino et al. showed that the expression of TGF-β1, BMP-4, and BMP-7 increases after post-irradiation day 4, followed by enhancement of calcification [36].

The effect of PBMT on the differentiation and dynamics of MSCs and other cells involved in bone tissue formation have been studied in recent years. Regarding the relationship between PBMT and osteoclasts, Bouvet-Gerbettaz et al. [37] showed that PBMT using a diode laser enhanced differentiation of irradiated murine MSCs to osteoclasts and calcification. In addition, Shirazi et al. [38] observed a significant increase in the amount of tooth movement and the number of osteoclasts present on the surface of the alveolar bone on the pressure side of tooth movement due to PBMT. As a result, bone resorption on the pressure side was promoted and bone remodeling was activated. This may be due to the promotion of osteoclast differentiation by PBMT, but this conclusion is speculative because there are relatively few in vitro studies of osteoclasts compared with osteoblasts. In contrast, Hirata et al. [39] showed that PBMT with irradiation on cultured MSCs increased expression of molecules essential for differentiation into osteoblasts (e.g., Cbfa1, osterix, and BMP) and induced bone formation. Ozawa et al. [40] showed that laser irradiation of osteoprogenitor cells isolated from rat palatine bone increased cell proliferation and ALP activity, thereby enhancing bone formation. Naka and Yokose showed that PBMT of tibial bone defects through the skin using a CO_2_ laser resulted in differentiation of osteoprogenitor cells into osteoblasts in the outer layer of the cortical bone, thereby accelerating bone formation [41]. These studies show that treatment of bone (including extraction sockets) using diode and CO_2_ lasers have similarities in healing morphology and cell dynamics.

## 8. Effect of PBMT on MSCs Transplantation in Regenerative Medicine

Recent studies have aimed to develop new wound healing and tissue-activating therapies based on the enhancing effect of PBMT on differentiation, activation, and calcification of MSCs and cells associated with bone formation [42,43,44]. Other studies have investigated the effect of applying PBMT to MSCs in regenerative medicine in order to induce regeneration or repair of non-bone tissues (including heart tissue, blood vessels, and nerves) [45,46].

In such treatments, bone marrow was first collected by bone marrow aspiration, and MSCs were isolated, cultured, transplanted to the damaged tissue, and then stimulated by PBMT. In a study by Nagata et al., cultured MSCs were transplanted into bone defects artificially created in rat skulls, and immunohistology showed that the expression levels of Runx 2 (a decisive transcription factor for differentiation from undifferentiated mesenchymal cells to osteoblasts) and a bone matrix protein osteocalcin were significantly increased in a PBMT-treated group compared with a control group, and histological and morphological examination showed significant increases in the volume of bone formed and the rate of bone formation in the PBMT-treated group compared with the control group [42]. Bayat and Jalalifirouzkouhi reported that the structure of osteoporotic bone tissue was improved by transplantation of PBMT-stimulated cultured MSCs [43]. Fekrazad et al. performed an experiment in which rat bone marrow was aspirated and BMSCs were isolated, cultured in a monolayer, and implanted into a full-thickness osteochondral defect (4 mm in diameter) that had been created in the patellar groove of both knees in the same rabbits. Then one knee was subjected to diode laser irradiation (810 nm; energy density, 4 J/cm^2^) every other day for 3 weeks. Improved healing in osteochondral defects was seen for the combination of BMSCs and LLLT compared with BMSCs alone—this improvement was mainly attributable to the formation of new bone rather than new cartilage. [44]. Cultured MSCs have also been transplanted to non-bone tissues. In a study by Yang et al., cultured MSCs were transplanted to the crushed sciatic nerve and significant changes were demonstrated when PBMT was performed. These changes included improved repair of the damaged nerves and reduced inflammatory cells (shown by histology), improvement in walking (assessed using the sciatic functional index), and improvement in electrophysiological function [45]. In a study using a mouse model of Alzheimer’s disease, Oron and Oron reported that PBMT of MSCs increased their ability to mature toward a monocyte lineage, and increased phagocytosis of soluble amyloid b, thereby improving cognition and spatial learning, and reducing the risk of dementia [46,47]. Tuby et al. performed transplantation of MSCs into the infarcted rat heart and showed a 53% reduction in infarct size, significant increases in angiogenesis (1.4-fold higher) and vascular endothelial growth factor (2-fold higher), a significant decrease in myocardial fibrosis, and recovery of heart function when PBMT was performed after implantation [48].

Many more studies have used MSCs toward the establishment of regenerative medicine, and it is becoming clear that PBMT plays an important role as adjuvant therapy in tissue regeneration. Although chemicals or gene recombination using viruses is commonly used to preserve the undifferentiated status or induce differentiation of stem cells, clinically adequate results have not yet been obtained. On the other hand, transformation of cultured stem cells into cancer cells after transplantation has been reported [49]. Therefore, there remains a pressing need to establish the clinical application of PBMT for MSCs.

At present, the mainstream idea is that the presence of cytochrome c oxidase is essential for cell activity induced by PBMT irradiation. Cytochrome c oxidase is likely involved in the promotion of wound healing by PBMT. In this regard, diode laser light in the red or near-infrared region, with wavelengths of 600–700 nm and 780–1100 nm, respectively, interacts with cytochrome c oxidase to stimulate the mitochondrial electron transport chain and increase ATP production, resulting in enhancement of tissue repair and healing through activation of stem cells [50,51,52]. However, the effect of CO_2_ laser light on cytochrome c oxidase has not yet been reported, and the enhancement of wound healing and cell differentiation by PBMT using a CO_2_ laser are not as well understood as those by PBMT using a diode laser. For this reason, basic research on PBMT using a CO_2_ laser both in vivo and in vitro, which is currently extremely limited, needs to be performed to establish evidence-based medicine.

## 9. Effect of Laser Irradiation on Dental Pulp Stem Cells

Dental pulp stem cells (PDSCs), which have a higher self-replication ability and similar potency compared with MSCs, are of great interest, and have been shown to be capable of differentiating into osteoblasts, chondrocytes, nerve cells, and odontoblasts [53,54,55,56,57]. While collection of MSCs requires bone marrow aspiration, PDSCs are isolated from extracted deciduous teeth or third molars, or teeth extracted for orthodontic treatment, and thus, an adequate number of cells can be obtained in a noninvasive and relatively simple manner. It is also worth noting that MSCs are present in a trace amount of 0.001% to 0.01% of the cell components in bone marrow [53,54]. Dental pulp is protected by enamel and dentin hard tissue, which has the advantage that PDSCs are less susceptible to cumulative genetic and environmental effects [58].

PDSCs are collected by the isolation and culture method originally devised by Gronthos and colleagues in 2000 [53,54], and basic research to establish regenerative medicine for transplantation of PDSCs into damaged or injured tissue is ongoing. In particular, it has been widely applied to bone [59,60], cartilage [61,62], and nerves [63], and good results have been reported.

Recently, the number of research reports on the effect of PBMT on PDSCs as well as MSCs has been gradually increasing. A 600-nm diode laser is usually used for PBMT of DPSCs, as used for PBMT of MSCs and other cells, so that laser light is absorbed by mitochondria, thereby activating cells and tissues. However, the timing of responses and the interactions that occur after PBMT in the host tissue remain largely unclear. Toward understanding these aspects, many basic and clinical studies with different focuses have been reported.

The effect of PBMT on DPSCs has often been judged in terms of cell growth, survival rate, and cellular activities. Eduardo et al., Holder et al., and Zaccara et al. reported that PBMT irradiation of DPSCs using an InGaAlP diode laser (wavelength, 600 nm) or an LED laser (wavelength, 635 nm) was likely to increase cell growth and survival, as shown by BrdU immunofluorescence staining and MTT assay, and also increased ATP production and mitochondrial metabolic activity, as well as cell activity based on NO level measurement [64,65,66]. Moreover, Ferreira et al. reported that DPSCs were able to remain undifferentiated and replicate for a short period after PBMT [67], while a review by Marques et al. indicated that PBMT had no deleterious effects on human DPSCs, although no clear conclusion was reached on that [68]. Farahani reported that PBMT may increase the proliferation of DPSCs, but the optimal PBMT conditions (e.g., wavelength, energy density, output, and duration of irradiation) need to be determined in future studies [69].

With respect to the effect of PBMT on the mineralization ability of human DPSCs, Matsui et al. irradiated dental pulp cells in vitro with a diode laser at 0.5 W for 500 s and 1.0 W for 500 s, and found that the number of calcified nodules, calcium production, ALP activity, and expression of BMP and osteocalcin were significantly higher with 1.0 W irradiation in a irradiation time-dependent manner [70,71]. Ohbayashi et al. showed similar results, although a different irradiation duration was used [72].

PBMT-stimulated human DPSCs have also been tested in other tissues, but basic research is limited. In the future, they need to be compared with PBMT-stimulated MSCs, which have been well studied in both and clinical research.

Taken together, basic cytological and histological research is needed to investigate the effect of PBMT on DPSCs. Nevertheless, results of currently available clinical studies strongly suggest a favorable effect of PBMT on cell differentiation, proliferation, and activity, as well as on subsequent tissue activation and wound healing.

## 10. Conclusions

As shown by the example of wound healing of the rat extraction socket by PBMT presented here, PBMT applying a CO_2_ laser or diode laser to extraction wounds is a safe and simple technique for effectively enhancing wound healing and socket preservation. Of the two types of laser devices shown above, the diode laser has been investigated in many basic studies on wound healing, and thus the mechanism is being clarified. However, further basic research is needed to investigate how PBMT using a CO_2_ laser enhances the ability of tissues to repair and heal themselves through enhancement of activation and differentiation of MSCs. However, PBMT for transplanted MSCs and DPSCs using a CO_2_ laser or diode laser appears to be a safe and simple method for increasing cell differentiation, proliferation, and activity and thereby promoting tissue activation and wound healing.

## Figures and Tables

**Figure 1 ijms-21-06850-f001:**
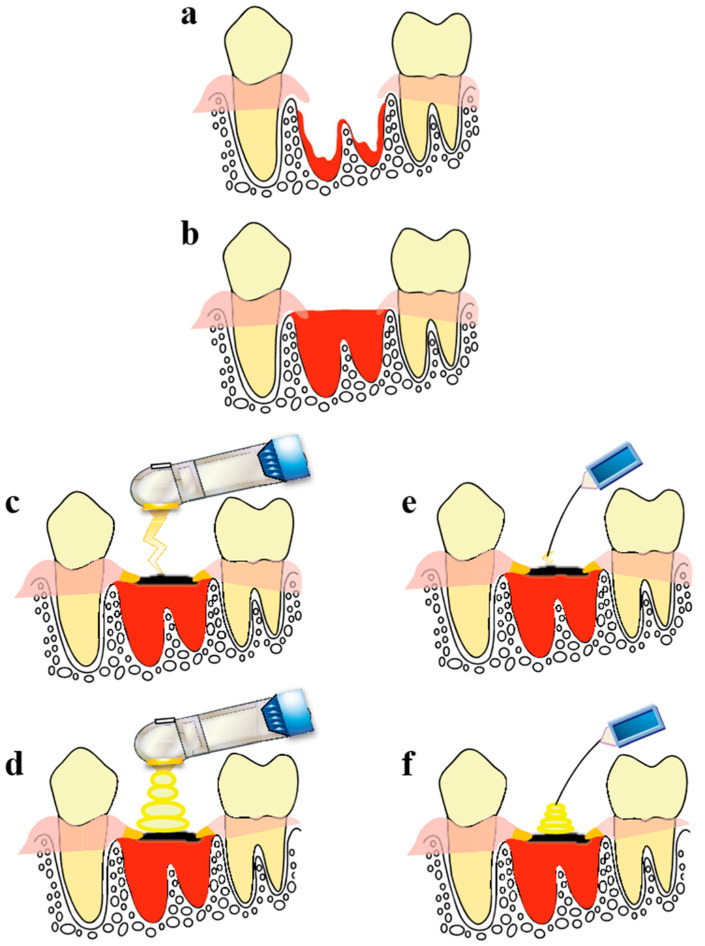
Laser treatment protocol. (**a**) Extraction site immediately after tooth extraction. (**b**) The inside of the tooth extraction socket was allowed to fill with an adequate amount of blood. (**c**,**e**) Carbonization of the blood in the superficial layer of the extraction socket with high-intensity laser therapy (HILT) to create an artificial eschar followed by fusion of the artificial eschar to the gingiva surrounding the extraction socket to prevent it from falling off. (**d**,**f**) Photobiomodulation therapy (PBMT) is performed the day after extraction to enhance healing. The procedure using the CO_2_ laser is shown in (**c**,**d**), and the procedure using the diode laser is shown in (**e**,**f**). Reproduced from [12] under Creative Commons CC BY-NC 4.0.

**Figure 2 ijms-21-06850-f002:**
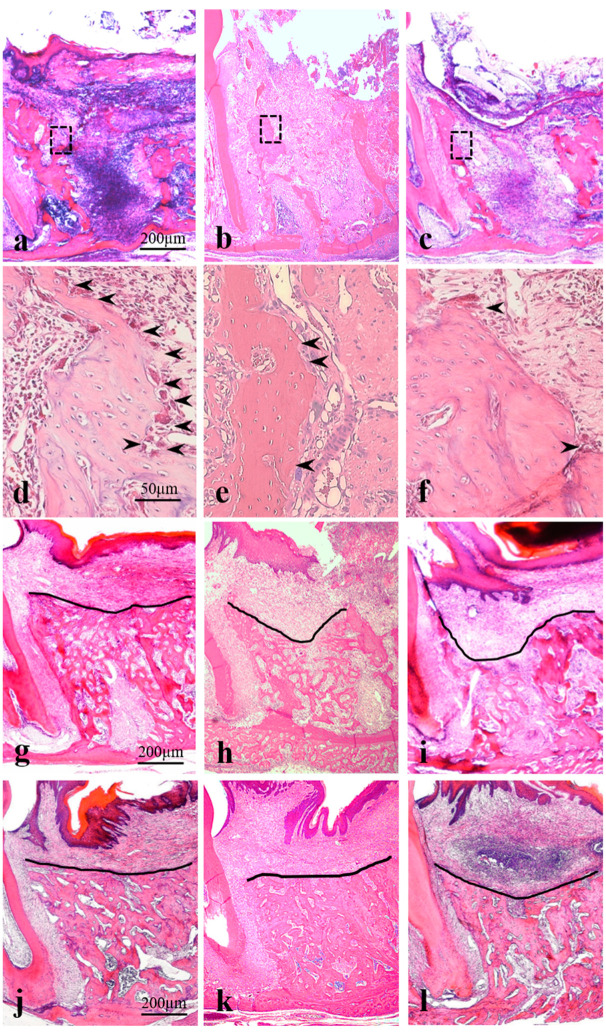
Histopathology on post-extraction days 3, 7, and 21. H&E staining; (**a**–**c**,**g**–**l**) Entire extraction socket, original magnification ×40. (**d**–**f**) Enlargement of areas indicated by the dotted line in (**a**–**c**), original magnification ×100. (**a**,**d**,**g**,**j**) CO_2_ group. (**b**,**e**,**h**,**k**) Diode group. (**c**,**f**,**i**,**l**) Control group. (**a**–**f**) Post-extraction day 3. (**g**,**h**) Post-extraction day 7. (**j**–**l**) Post-extraction day 21. Arrowheads (➤) indicate osteoclast-like cells. Solid lines indicate bone level. (**a**,**b**) Progressive organization surrounding the tooth extraction socket, with blood clots observed in only the central area. (**c**) The inside of the tooth extraction socket is almost completely filled with blood clots. (**d**) Many osteoclast-like cells are present in the alveolar bone wall in the extraction socket, showing active bone resorption. (**e**) Near absence of osteoclast-like cells with rapid organization. (**f**) Only a few osteoclast-like cells are present in the alveolar bone wall in the extraction socket. (**g**) Bridging osteoneogenesis extending from the superficial to middle layers of the tooth extraction socket and the bone trabeculae width are favorable, and the structures are more densely packed. (**h**) New bone formation seen from the fundus to the shallow layer of the extraction socket. (**i**) Cancellous bone formation in the extraction socket is immature, weak, and not continuous. (**j**,**k**) Flattened alveolar crest with almost no concavity observed. (**l**) There is a dish-shaped concavity on the surface of the alveolar crest, and the cancellous bone is less dense. Reproduced from [12] under Creative Commons CC BY-NC 4.0.

**Figure 3 ijms-21-06850-f003:**
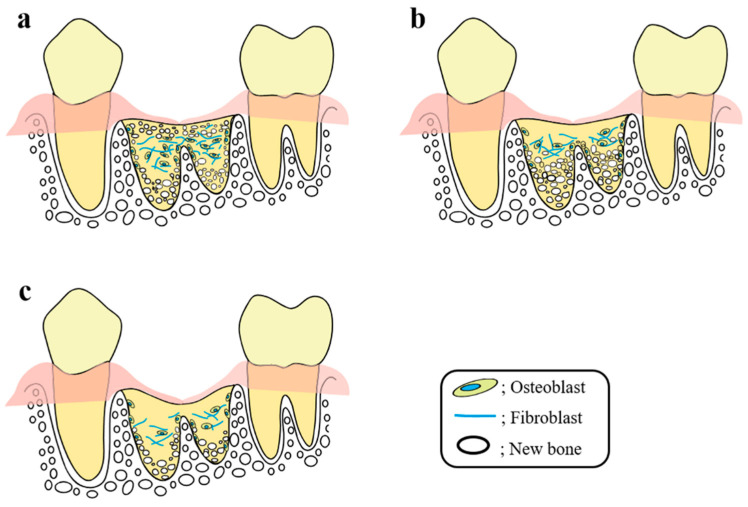
Differences in osteoneogenesis in tooth extraction sockets between the (**a**) CO_2_ group, (**b**) diode group, and (**c**) control group. (**a**) Early in the healing process after tooth extraction, laser stimulation causes many osteoclast-like cells to appear and active bone resorption to occur from the shallow to middle layers of the extraction socket wall (activation of bone remodeling). Next, the stimulated osteoblasts produce collagen fibers associated with bone formation. These fibers form a scaffold, along which the osteoblasts migrate to form new bone. The formation of new bone is cross-linked with the collagen fiber scaffolding observed at the same site. Formation of the bone lining under the mucosa of the extraction wound prevents the depression of the mucosal epithelium and preserves the alveolar crest height. (**b**) Laser-stimulated osteoblasts present along the whole circumference of the socket produce collagen to fill the entire socket, and organization occurs at a considerably earlier stage compared with the CO_2_ group. Then, fibrous granulation tissue is formed, and osteoblasts migrate to the collagen fiber to replace the granulation tissue with bone tissue. As a result, depression of the mucosal epithelium is prevented, and the alveolar crest height is preserved. (**c**) Few osteoclast-like cells are seen in the wall of the extraction socket during the early stage of healing. New bone formation in the extraction socket begins at the bottom of the extraction socket. The appearance of cells involved in bone formation is delayed compared with the laser treatment groups, and new bone formation is also delayed. As a result, the mucosal epithelium around the extraction wound invades the extraction socket and a concavity forms at the center of the alveolar crest with a corresponding decrease in alveolar crest height. Reproduced from [12] under Creative Commons CC BY-NC 4.0.

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
