# Peer review of "Wound Healing and Cell Dynamics Including Mesenchymal and Dental Pulp Stem Cells Induced by Photobiomodulation Therapy: An Example of Socket-Preserving Effects after Tooth Extraction in Rats and a Literature Review"

_ijms, 2020, doi:10.3390/ijms21186850_

Round 1

Reviewer 1 Report

The is review addresses the effects and mechanisms of HILT and PBMT (diode lasers and CO2 respectively) on wound healing after tooth extraction and performs a small study on this matter. Over all, this a well written review especial for the review part. However, I do have some comments and suggestions for the animal experiment for the authors to consider revising this manuscript .

1: In the animal experiment, how many rats were used in each group and at each time points?

2: Is the animal experiment approved by ethics committee?

3: The authors claim that “the effects of PBMT on the wound healing of bone tissues including the extraction socket are reviewed histologically, biochemically, and cytologically” in the abstract. It seems only the HE staining is done. What are biochemical or cytological tests performed?

4: Results showed in Fig 2 should be quantified especial osteoclast-like cells. Recommend to use a special marker to stain the tissue by IHC then quantify.

5: To better support the conclusion of this animal study, collagen staining in the healing/healed socket is highly recommended.

Author Response

Response to Reviewer 1

1: In the animal experiment, how many rats were used in each group and at each time points?

Response: In response to your question, we have added the following text. “A total of 54 rats, 6 for each observation time point, were examined in this study.” (page 4, line 122).

2: Is the animal experiment approved by ethics committee?

Response: Yes, it was approved. We have noted this in the text. “The study protocol complied with the Osaka Dental University Guidelines for Animal Experiments (approval no. 18-01008).” (page 4, lines 153-154).

3: The authors claim that “the effects of PBMT on the wound healing of bone tissues including the extraction socket are reviewed histologically, biochemically, and cytologically” in the abstract. It seems only the HE staining is done. What are biochemical or cytological tests performed?

Response: We apologize for the misleading text. To clarify this point, we have revised the text as follows.

“Then, the effects of PBMT on the wound healing of bone tissues are reviewed from histological, biochemical, and cytological perspectives based on our own study of the extraction socket as well as studies by other researchers.”

*Revised text is highlighted in gray.

4: Results showed in Fig 2 should be quantified especial osteoclast-like cells. Recommend to use a special marker to stain the tissue by IHC then quantify.

Response: Thank you for your suggestion.

The following explains the method for observing osteoclasts in this review.

The osteoclasts targeted in this study represent active multinucleated giant cells that have formed bone resorption lacunae on the surface of the tooth extraction socket wall.

Certainly, as you pointed out, if you want to observe and quantify osteoclasts, immunostaining typified by TRACP-5b would be the best choice.

However, because TRACP-5b is an enzyme in osteoclasts, quantification would not be accurate for our purposes because it also stains inactive osteoclasts that do not form bone resorption lacunae and osteoclasts buried in bone tissue.

Furthermore, because TRACP-5b is an enzyme in osteoclasts, if the enzyme is overly abundant in the surrounding tissue, they will also be stained, making observation difficult.

In contrast, HE staining is suitable for observing bone remodeling, including changes in cells and tissues around the bone resorption lacunae.

 This is because we can observe osteoclasts and bone resorption lacunae as a set, that is, as activated osteoclasts, and thus we believe that sufficient quantification is possible.

5: To better support the conclusion of this animal study, collagen staining in the healing/healed socket is highly recommended.

Response: Thank you for your suggestion.

The staining of collagen fibers you asked is explained below.

We believe that immunostaining of type I collagen fibers is an excellent method for verifying and quantifying the morphology and structure of wound healing in the extraction socket.

In this study, the healing morphology in the tooth extraction socket has been verified for the purpose of observing the socket preservation effect by laser irradiation after tooth extraction.

This is because there is no basic research on the healing pattern after laser irradiation, despite the FDA's recommendation for this procedure.

Therefore, compared with staining to quantify collagen fibers, HE staining to observe the healing morphology histopathologically can show not only the morphology of new bone formation but also the expression and migration of cells involved in bone formation.

 Furthermore, it is possible to distinguish the boundary between the wall of the extraction socket and the new bone in the extraction socket in the HE-stained specimen.

If the specimen is not stained in this way, it may be difficult to distinguish the boundary between new bone and mature alveolar bone, and it is therefore impossible to accurately measure the amount of new bone formation.

Reviewer 2 Report

In the draft of review article, the authors discussed the applications of photobiomodulation therapy in socket preservation, mainly based on their previous studies (ref 14 and 15). It is recommended to highlight/compare the findings of ref 14/15 with other literature. Other issues that need to be clarified/addressed are outlined below.

Title

It is recommended to highlight how to “enrich” the NRL film in the title. The elements red propolis, silver nanoparticle and plasma treatment shall be included.

The authors proposed NRL film as “occlusive wound healing device”. However, there is no exp related to occlusive wound. It is difficult to understand the reason why they highlighted “occlusive wound healing” in title.

Abstract,

Page 1, line 20-21,

“basic research has revealed that cell differentiation, proliferation, and activity and subsequent tissue activation.”

Various types of cells are involved in wound healing. Otherwise, different cells shall be active at different phases. The authors shall clarify the what regulation of PBMT to specific cells (for example, epithelial cell, fibroblast or osteoblast etc).

Keywords

“Low-level laser therapy” is not a critical point in this review. It is recommended to be removed (may be confused to PBMT).

Introduction

Page 1, line 41-43,

“photochemical actions that stimulate tissues to induce an anti-inflammatory response and wound healing [1-5],”

Ref 2 is related to MSCs in articular cartilage, and Ref 3 is demonstrated the effect of photobiomodulation associated with vitamin A. It is recommended to cite references related to this article.

Page 2, line 45-46,

“We have investigated wound healing in the extraction socket (a recess formed after tooth extraction) following laser irradiation [14,15].”

The authors shall briefly introduce the results of their studies (the authors had introduced the results of these 2 ref in section 4; therefore, the authors may also revise above sentence).

Page 2, paragraph 2,

The authors may provide more information regarding the reducing height in alveolar crest, and the requirement of alveolar ridge for dental implant.

Page 2, paragraph 3,

Several concerns of grafting materials have been discussed here. A reference is necessary to support these descriptions such as “residual graft materials may remain and not be replaced by bone tissue” since it shall be a time-dependent process.

Page 2, section 2,

The authors mentioned the “longest wavelength” and relatively short wavelength of dental lasers. It is recommended to provide the wavelengths of other dental lasers, which can help readers to verify the long or short wavelength of dental lasers (for example, 2940 nm for Er:YAG).

Page 4, section 4,

Since this draft is a review article, it is not necessary to repeat the details for exp (such as animal species, preparation for histological examination etc). The authors shall highlight on their findings but not the procedures. Especially, both Fig. 1 and Fig. 2 are cited from ref 14.

Page 7, section 5,

The authors compared the outcomes of laser treatments with other conventional approaches for socker preservation. However, comparisons for some widely methods shall also be included (GTR, PRF, etc).

Page 8, line 216,

“many osteoclast-like cells were observed in the relatively shallow layers”

Fig. 2 represented general H&E staining. It is difficult to verify “osteoclast-like cells”. Additional descriptions/explanations shall be provided.

Page 9, line 294-297,

“Tang and Chai showed that PBMT using a CO2 laser on experimental radial fractures in rats resulted in the appearance of many osteoclasts, increased blood supply due to increased capillary formation, and accelerated calcification at the early stage of treatment.”

Osteoclasts also has an important role in bony tissue remodeling. It is highly recommended to discuss the effects of PBMT on osteoclasts in detail. The molecular mechanism shall be discussed. In addition, is there any finding related to osteoclasts in ref 14 and 15?

Page 11, paragraph 2,

The correlations among PBMT, cytochrome c oxidase, and wound healing shall be addressed. Is PBMT stimulating cytochrome c oxidase expression? Or PBMT enhancing cytochrome c oxidase function?

In line 363, the targeting cell is stem cells. Is the finding also noticed in mature osteoblasts?

For this review article, the authors focused on the effect of dental lasers on osteoblasts and MSCs (DPSCs). How about the effect of PBMT on reepithelialization?

The limitations of CO2 and diode lasers shall be discussed.

Author Response

Response to Reviewer ï¼’

Comments and Suggestions for Authors

In the draft of review article, the authors discussed the applications of photobiomodulation therapy in socket preservation, mainly based on their previous studies (ref 14 and 15). It is recommended to highlight/compare the findings of ref 14/15 with other literature. Other issues that need to be clarified/addressed are outlined below.

Response: We previously reported the socket preservation effect of the carbon dioxide laser, the healing morphology of which is shown in Fig. 2.

No other studies have performed carbon-dioxide-laser irradiation after tooth extraction.

References 24-26 (lines 225-232) provide similar descriptions of this healing process, particularly Ref. 26.

This means that filling the extraction socket with hyaluronic acid immediately after tooth extraction led to a large number of osteoclasts early in the wound healing process, and that final healing was also promoted (line 228-232).

Regarding the socket preservation effect from using our semiconductor laser, there are no particular differences from what has been reported by other researchers, and the histology is quite similar.

However, in our study, we reported that not only was wound healing promoted, but there was almost no concavity in the mucosa of the tooth extraction wound and the bone level was well maintained; no other studies have reported such findings.

From the above, we can say that not only did our approach promote wound healing by laser irradiation after tooth extraction, but it was also shown to be safer than the conventional method of socket preservation (lines 60-67) for maintaining the alveolar bone level. Therefore, we have shown this approach to be a simple and risk-free method.

To induce such a healing morphology, activation of tissues, including cell activity and differentiation by PBMT, is important.

This is shown in this review.

 Title

It is recommended to highlight how to “enrich” the NRL film in the title. The elements red propolis, silver nanoparticle and plasma treatment shall be included.

The authors proposed NRL film as “occlusive wound healing device”. However, there is no exp related to occlusive wound. It is difficult to understand the reason why they highlighted “occlusive wound healing” in title.

Response: Thank you for your question.

The word "occlusive" in "occlusive wound healing device" is not used, but we double-checked that it was not included in the text by mistake. We are uncertain how to respond to this comment because the terms “NRL film” and “occlusive” do not appear anywhere in our paper.

Abstract,

Page 1, line 20-21,

“basic research has revealed that cell differentiation, proliferation, and activity and subsequent tissue activation.”

Various types of cells are involved in wound healing. Otherwise, different cells shall be active at different phases. The authors shall clarify the what regulation of PBMT to specific cells (for example, epithelial cell, fibroblast or osteoblast etc).

Response: Thank you for your comment.

We would like to contribute to the elucidation of PBMT irradiation for osteoclasts and osteoblasts involved in bone remodeling, including the extraction socket.

Therefore, we examined why PBMT increased the number of osteoclasts, whether the increased differentiation into osteoclasts was due to PBMT or whether osteoclasts were activated by PBMT.

In addition, it is necessary to verify whether PBMT increased the expression of osteoblasts, activated osteoblasts, or promoted the formation of new bone.

In particular, we think that it is important to conduct basic research to address the lack of reports on CO2 lasers.

Keywords

“Low-level laser therapy” is not a critical point in this review. It is recommended to be removed (may be confused to PBMT).

Response: In accordance with your suggestion, we have deleted this part.

Introduction

Page 1, line 41-43,

“photochemical actions that stimulate tissues to induce an anti-inflammatory response and wound healing [1-5],” Ref 2 is related to MSCs in articular cartilage, and Ref 3 is demonstrated the effect of photobiomodulation associated with vitamin A. It is recommended to cite references related to this article.

Response: Thank you for your advice.

I changed the location where ref 2 is quoted and cited it as ref 44.

The following text referring to ref 44 was added to lines 362-369 on page 11.

“Fekrazad et al. performed an experiment in which rat bone marrow was aspirated and BMSCs were isolated, cultured in a monolayer, and implanted into a full-thickness osteochondral defect (4 mm in diameter) that had been created in the patellar groove of both knees in the same rabbits. Then one knee was subjected to diode laser irradiation (810 nm; energy density, 4 J/cm2) every other day for 3 weeks. Improved healing in osteochondral defects was seen for the combination of BMSCs and LLLT compared with BMSCs alone, this improvement was mainly attributable to the formation of new bone rather than new cartilage. [44].”

Ref 3 was deleted because it was difficult to cite it because of the strong effect of vitamin A.

Page 2, line 45-46,

“We have investigated wound healing in the extraction socket (a recess formed after tooth extraction) following laser irradiation [14,15].”

The authors shall briefly introduce the results of their studies (the authors had introduced the results of these 2 ref in section 4; therefore, the authors may also revise above sentence).

Response:

In accordance with your comment, we have moved the abovementioned sentence to a new paragraph. We have also added a sentence to explain our research results later.

In addition, an introductory sentence was added to explain the healing process of tooth extraction sockets. (lines 46-49)

Page 2, paragraph 2,

The authors may provide more information regarding the reducing height in alveolar crest, and the requirement of alveolar ridge for dental implant.

Response: In accordance with your suggestion, we have added the following text to lines 53-59.

“In particular, implant treatment is said to require an alveolar bone height of 10 mm or more and a width of 6 mm or more. However, these requirements depend on the implant type, length, width, and superstructure type. Furthermore, the quality of the alveolar bone varies widely from person to person, so it is not possible to specify the precise requirements. In addition, the requirements cannot be specified internationally because the guidelines differ by country. For that reason, preserving alveolar crest height, together with rapid wound healing after tooth extraction, is essential. This is the concept behind socket preservation.”

Page 2, paragraph 3,

Several concerns of grafting materials have been discussed here. A reference is necessary to support these descriptions such as “residual graft materials may remain and not be replaced by bone tissue” since it shall be a time-dependent process.

Response:

In accordance with your comment, we have added newly cited Reference 19(line 209).

“However, Chan et al. reported that residual (15%-36%) grafting materials such as Bio-Oss® Collagen and hydroxyapatite remained in granular form for up to 6 months after placement in the extraction socket [19].”

Page 2, section 2,

The authors mentioned the “longest wavelength” and relatively short wavelength of dental lasers. It is recommended to provide the wavelengths of other dental lasers, which can help readers to verify the long or short wavelength of dental lasers (for example, 2940 nm for Er:YAG).

Response: 

In accordance with your comment, we have added the following text (lines 78-80).

“For comparison, other lasers with shorter wavelengths include the Er:YAG laser (2940 nm), followed by the Nd:YAG laser (1064 nm), diode laser (600-1000 nm), and argon laser (488 nm).”

Page 4, section 4,

Since this draft is a review article, it is not necessary to repeat the details for exp (such as animal species, preparation for histological examination etc). The authors shall highlight on their findings but not the procedures. Especially, both Fig. 1 and Fig. 2 are cited from ref 14.

Response: Thank you for your comment. 

Regarding the description of the operative method in the text, the sentence that overlapped with the explanation in Figure 1 was deleted and it was described to Lines 125-126, as "Following this, the procedure was performed as shown in Figure 1."

However, the text regarding the results and discussion will remain as is.

Page 7, section 5,

The authors compared the outcomes of laser treatments with other conventional approaches for socker preservation. However, comparisons for some widely methods shall also be included (GTR, PRF, etc).

Response:

In accordance with your comments, we have added the following text (lines 211-221).

“The guided bone regeneration (GBR) method uses a membrane to prevent the invasion of the gingiva, that is, to secure a site for new bone formation by making space and promoting regeneration. In some cases, autologous bone, artificial bone, or a substitute material is transplanted. A systematic review of the effects of barrier membranes on bone formation reported an average increase in vertical bone formation of 0.32 mm, minimum postoperative infection, wound dehiscence, and exposure of membranes and implants [21]. However, Chen et al reported that immediate implant placement, which is performed simultaneously with the GBR method, may cause the membrane to rupture or be exposed [22]. The platelet-rich plasma method, which fills the extraction socket with the collected and separated autologous plasma, is also attracting increasing attention. However, because the substance to be filled is a blood component, it is necessary to mix it with an artificial material or shield it with a membrane, rather than using it alone [23].”

Page 8, line 216,

“many osteoclast-like cells were observed in the relatively shallow layers”

Fig. 2 represented general H&E staining. It is difficult to verify “osteoclast-like cells”. Additional descriptions/explanations shall be provided.

Response: Thank you for your suggestion.

The following explains the method for observing osteoclasts in this review.

The osteoclasts targeted in this study represent active multinucleated giant cells that have formed bone resorption lacunae on the surface of the tooth extraction socket wall.

Certainly, as you pointed out, if you want to observe and quantify osteoclasts, immunostaining typified by TRACP-5b would be the best choice.

However, because TRACP-5b is an enzyme in osteoclasts, quantification would not be accurate for our purposes because it also stains inactive osteoclasts that do not form bone resorption lacunae and osteoclasts buried in bone tissue.

Furthermore, because TRACP-5b is an enzyme in osteoclasts, if the enzyme is overly abundant in the surrounding tissue, they will also be stained, making observation difficult.

In contrast, HE staining is suitable for observing bone remodeling, including changes in cells and tissues around the bone resorption lacunae.

 This is because we can observe osteoclasts and bone resorption lacunae as a set, that is, as activated osteoclasts, and thus we believe that sufficient quantification is possible.

Regarding the observed osteoclasts, the following description has been added to section 4, lines 164-166.

“In this review, osteoclast-like cells were defined as a set of multinucleated giant cells present on the surface of the tooth extraction wall and the bone-resorbing cavities in contact with these cells.”

Page 9, line 294-297,

“Tang and Chai showed that PBMT using a CO2 laser on experimental radial fractures in rats resulted in the appearance of many osteoclasts, increased blood supply due to increased capillary formation, and accelerated calcification at the early stage of treatment.”

Osteoclasts also has an important role in bony tissue remodeling. It is highly recommended to discuss the effects of PBMT on osteoclasts in detail. The molecular mechanism shall be discussed.

Response: Thank you for your advice.

Most reports on PBMT and osteoclasts are related to orthodontic treatment.

Generally, the alveolar bone on the pressure side receives continuous pressure due to tooth movement.

As a result, in the initial stage of corrective movement, osteoclasts appear on the pressure side, alveolar bone resorption occurs, and the teeth move.

It has been reported that the combined use of PBMT with this treatment promotes tooth movement.

In in vivo studies using rats, the number of osteoclasts on the alveolar bone surface on the pressure side significantly increases in proportion to the irradiation dose of PBMT and the tooth movement also significantly increases.

In addition, it was reported that there was a significant increase in RANKL expressed by osteoblasts in the alveolar bone on the pressure side and an increase in ALP expression on the tension side.

That is, compared with the non-irradiated group, PBMT promotes osteoclast differentiation on the pressure side and also promotes bone resorption.

Also, most studies have reported that bone formation is significantly promoted by PBMT on the other tension side.

The overwhelming number of basic studies on PBMT and osteogenesis have involved in vitro studies on osteoblasts, but few such studies have examined osteoclasts.

Accordingly, it is worth mentioning the report by Tang and Chai.

From the above, it can be considered that the promotion of bone formation is due to the promotion of wound healing, that is, the promotion of osteoclast differentiation, based on the appearance of many osteoclasts during the initial stage of wound healing of the extraction socket by PBMT in this study.

Furthermore, from the report on PBMT and orthodontic treatment, PBMT increases osteoclasts and RANKL expression on the pressure side. These reports together with the significant expression of osteoclasts in the early stages of wound healing of tooth extraction wounds observed in this study can be expected to significantly promote bone remodeling.

However, because there are few in vitro studies of this phenomenon, the conclusion is speculative.

Therefore, we intend to investigate this further in a future study.

Based on the above contents, the following text was added to page 10, lines 332-337, and a reference was also added.

“In addition, Shirazi et al. [38] observed a significant increase in the amount of tooth movement and the number of osteoclasts present on the surface of the alveolar bone on the pressure side of tooth movement due to PBMT. As a result, bone resorption on the pressure side was promoted and bone remodeling was activated. This may be due to the promotion of osteoclast differentiation by PBMT, but this conclusion is speculative because there are relatively few in vitro studies of osteoclasts compared with osteoblasts. In contrast,”

In addition, is there any finding related to osteoclasts in ref 14 and 15?

Response: Thank you for your advice.

Regarding the findings regarding PBMT and osteoclasts, the same comments as in reference 14 are described on lines 234 to 236 in the text. Because there were few comments on the relationship between PBMT and osteoclasts, we have added it to lines 332-337 of the text, along with the answers to the previous questions, to supplement it.

Page 11, paragraph 2,

The correlations among PBMT, cytochrome c oxidase, and wound healing shall be addressed. Is PBMT stimulating cytochrome c oxidase expression? Or PBMT enhancing cytochrome c oxidase function?

Response: Thank you for your question.

PBMT stimulates the cytochrome C oxidase present in the inner mitochondrial membrane of cells. As a result, ATP production is increased and converted into energy.

Therefore, PBMT stimulates cytochrome C oxidase but is not known to enhance it.

In line 363, the targeting cell is stem cells. Is the finding also noticed in mature osteoblasts?

Response: Thank you for your question

In basic in vivo and in vitro studies, it has been gradually revealed that PBMT irradiation of undifferentiated cells, cells present in damaged tissues, and cultured cells exhibit differentiation, proliferation, and activity.

However, some of the abovementioned effects on cells are gradually being clarified depending on the type and characteristics of the laser device, whereas others remain to be clarified.

The distinction between mature and immature osteoblasts is not known, but PBMT is considered to activate mitochondria in osteoblasts.

However, no study has compared immature and mature osteoblasts, so it is not known what the effect of PBMT is on mature osteoblasts.

The effect of PBMT on cultured osteoblast cells is described in reference 30-36.

It is considered that osteoblasts cultured from mature osteoblasts have cell activity and functions such as increased alkaline phosphatase activity, increased type I collagen and osteopontin expression, and accelerated calcification of wound healing, as shown in the above reference.

For this review article, the authors focused on the effect of dental lasers on osteoblasts and MSCs (DPSCs). How about the effect of PBMT on reepithelialization?

Response:

At present, we are also examining the re-epithelialization of the mucosal epithelium of tooth extraction wounds.

In the present study, laser irradiation significantly reduced the appearance of myofibroblasts (which is related to epithelial scarring) present in the granulation tissue in the part that will become the mucosal epithelium of the tooth extraction in the future.

As a result, it is thought that this led to the suppression of the depression of the mucosal epithelium of the tooth extraction wound. (Of course, new bone formation in the extraction socket is one of the factors that suppresses the mucosal depression).

This phenomenon is not limited to tooth extraction wounds.

In recent years, it has been reported that laser irradiation of the wound surface after trauma and surgery, scarred skin tissue (burns, keloids), and the tongue dorsal significantly reduces expression of myofibroblasts and TGF-β1 (which is involved in myofibroblast differentiation). From the above, it can be concluded that laser irradiation in re-epithelialization is effective.

References

  • Tawfic SO, El-Tawdy A, Shalaby S, et al. Evaluation of Fractional CO2 Versus Long Pulsed Nd:YAG Lasers in Treatment of Hypertrophic Scars and Keloids: A Randomized Clinical Trial. Lasers Surg Med. 2020; in press.
  • Makboul M, Makboul R, Abdelhafez AH, Hassan SS, Youssif SM. Evaluation of the effect of fractional CO2 laser on histopathological picture and TGF-β1 expression in hypertrophic scar. J Cosmet Dermatol. 2014 Sep;13(3):169-79.
  • Pinheiro AL, Pozza DH, Oliveira MG, Weissmann R, Ramalho LM. Polarized light (400-2000nm) and non-ablative laser (685nm): a description of the wound healing process using immunohistochemical analysis. Photomed Laser Surg. 2005; 23(5): 485-492. de
  • Freitas AC, Pinheiro AL, de Oliveira MG, Ramalho LM. Assessment of the behavior of myofibroblasts on scalpel and CO(2) laser wounds: an immunohistochemical study in rats. J Clin Laser Med Surg. 2002; 20(4): 221-225.

The limitations of CO2 and diode lasers shall be discussed.

Response:

Laser treatment is basically considered not to have side effects like those of drugs.

However, there can be errors due to technical problems resulting from incorrectly set irradiation conditions and mistakes in the irradiation method.

In addition, electricity is used to generate semiconductor lasers, so caution is required for people who have pacemakers or artificial valves implanted in their bodies.

Taking laser irradiation of the extraction socket as an example, it can be applied to nearly every patient.

This method would be particularly effective for patients taking anticoagulants or antithrombotic agents or those prone to bleeding.

Round 2

Reviewer 1 Report

The authors have answered all my questions. 

Author Response

Comment: The authors have answered all my questions.
Response: Thank you for reviewing our manuscript and providing valuable feedback.

Reviewer 2 Report

The draft is well revised. One question still needs to be answered.

The possible disadvantages/side effects/limitations of CO2 and diode lasers in wound healing/ socket-preserving shall be discussed.

Author Response

The draft is well revised. One question still needs to be answered.

The possible disadvantages/side effects/limitations of CO2 and diode lasers in wound healing/ socket-preserving shall be discussed.

Response

In accordance with your comment, we have added text on page 9, starting on line 287, as follows.

Regarding potential problems in socket preservation using CO2 laser and diode laser irradiation, we consider there to be basically no side effects like those of drugs. However, errors may arise due to incorrectly set irradiation conditions or mistakes in the irradiation method. In particular, when using a diode laser, there is a high risk of the laser tip coming into contact with the bone surface because the laser tip must be brought into contact with the blood. As a result, there is a risk of burns and severe pain in the bone tissue, or worse, osteonecrosis. In addition, electricity is used to generate diode lasers, so caution is required for patients with pacemakers or artificial valves. In contrast, the laser tip of a CO2 laser does not require contact with the blood, and thus there is little or no effect on bone tissue, making it the safer option.

Laser irradiation of the extraction socket can be applied to nearly every patient. This method would be particularly effective in patients taking anticoagulants or antithrombotic agents or those prone to bleeding.